# Neural Architecture Retrieval

**Xiaohuan Pei**
Department of Computer Science
The University of Sydney, Australia
`xpei8318@uni.sydney.edu.au`

**Yanxi Li**
Department of Computer Science
The University of Sydney, Australia
`yanli0722@uni.sydney.edu.au`

**Minjing Dong**
Department of Computer Science
City University of Hong Kong, China
`minjdong@cityu.edu.hk`

**Chang Xu**
Department of Computer Science
The University of Sydney, Australia
`c.xu@sydney.edu.au`

## Abstract

With the increasing number of new neural architecture designs and substantial existing neural architectures, it becomes difficult for the researchers to situate their contributions compared with existing neural architectures or establish the connections between their designs and other relevant ones. To discover similar neural architectures in an efficient and automatic manner, we define a new problem Neural Architecture Retrieval which retrieves a set of existing neural architectures which have similar designs to the query neural architecture. Existing graph pre-training strategies cannot address the computational graph in neural architectures due to the graph size and motifs. To fulfill this potential, we propose to divide the graph into motifs which are used to rebuild the macro graph to tackle these issues, and introduce multi-level contrastive learning to achieve accurate graph representation learning. Extensive evaluations on both human-designed and synthesized neural architectures demonstrate the superiority of our algorithm. Such a dataset which contains 12k real-world network architectures, as well as their embedding, is built for neural architecture retrieval. Our project is available at www.terrypei.com/nn-retrieval.

## 1 Introduction

Deep Neural Networks (DNNs) have proven their dominance in the field of computer vision tasks, including image classification (He et al., 2016; Zagoruyko & Komodakis, 2016; Liu et al., 2021; Dong et al., 2021; Wang et al., 2018), object detection (Tian et al., 2019; Carion et al., 2020; Tan et al., 2020), etc. Architecture designs play an important role in this success since each innovative and advanced architecture design always lead to a boost of network performance in various tasks. For example, the ResNet family is introduced to make it possible to train extremely deep networks via residual connections (He et al., 2016), and the Vision Transformer (ViT) family proposes to split the images into patches and utilize multi-head self-attentions for feature extraction, which shows superiority over Convolutional Neural Networks (CNNs) in some tasks (Dosovitskiy et al., 2020). With the increasing efforts in architecture designs, an enormous number of neural architectures have been introduced and open-sourced, which are available on various platforms [1].

Information Retrieval (IR) plays an important role in knowledge management due to its ability to store, retrieve, and maintain information. With access to such a large number of neural architectures on various tasks, it is natural to look for a retrieval system which maintains and utilizes these valuable neural architecture designs. Given a query, the users can find useful information, such as relevant architecture designs, within massive data resources and rank the results by relevance in low latency. To the best of our knowledge, this is the first work to setup the retrieval system for neural architectures. We define this new problem as *Neural Architecture Retrieval* (NAR), which returns a set of similar neural architectures given a query neural architecture. NAR aims at maintaining both

---

[1] https://huggingface.co/, https://pytorch.org/hub/

existing and potential neural architecture designs, and achieving efficient and accurate retrieval, with which the researchers can easily identify the uniqueness of a new architecture design or check the existing modifications on a specific neural architecture.

Embedding-based models which jointly embed documents and queries in the same embedding space for similarity measurement are widely adopted in retrieval algorithms (Huang et al., 2020; Chang et al., 2020). With accurate embedding of all candidate documents, the results can be efficiently computed via nearest neighbor search algorithms in the embedding space. At first glance of NAR, it is easy to come up with the graph pre-training strategies via Graph Neural Networks (GNNs) since the computational graphs of networks can be easily derived to represent the neural architectures. However, existing graph pre-training strategies cannot achieve effective learning of graph embedding directly due to the characteristic of neural architectures. One concern lies in the dramatically varied graph sizes among different neural architectures, such as LeNet-5 versus ViT-L. Another concern lies in the motifs in neural architectures. Besides the entire graph, the motifs in neural architectures are another essential component to be considered in similarity measurement. For example, ResNet-50 and ResNet-101 are different in graph-level, however, their block designs are exactly the same. Thus, it is difficult for existing algorithms to learn graph embedding effectively.

In this work, we introduce a new framework to learn accurate graph embedding especially for neural architectures to tackle NAR problem. To address the graph size and motifs issues, we propose to split the graph into several motifs and rebuild the graph by treating motifs as nodes in a macro graph to reduce the graph size as well as take motifs into consideration. Specifically, we introduce a new motifs sampling strategy which encodes the neighbours of nodes to expand the receptive field of motifs in the graph to convert the graph to an encoded sequence, and the motifs can be derived by discovering the frequent subsequences. To achieve accurate graph embedding learning which can be easily generalized to potential unknown neural architectures, we introduce motifs-level and graph-level pre-train tasks. We include both human-designed neural architectures and those from NAS search space as datasets to verify the effectiveness of proposed algorithm. For real-world neural architectures, we build a dataset with 12k different architectures collected from Hugging Face and PyTorch Hub, where each architecture is associated with an embedding for relevance computation.

Our contributions can be summarized as: **1.** A new problem Neural Architecture Retrieval which benefits the community of architecture designing. **2.** A novel graph representation learning algorithm to tackle the challenging NAR problem. **3.** Sufficient experiments on the neural architectures from both real-world ones collected from various platforms and synthesized ones from NAS search space, and our proposed algorithm shows superiority over other baselines. **4.** A new dataset of 12k real-world neural architectures with their corresponding embedding.

## 2 RELATED WORK

**Human-designed Architecture**    Researchers have proposed various architectures for improved performance on various tasks (Li et al., 2022a; Dong et al., 2023). GoogLeNet uses inception modules for feature scaling (Szegedy et al., 2015). ResNet employs skip connections (He et al., 2016), DenseNet connects all layers within blocks (Huang et al., 2017), and SENet uses squeeze-and-excitation blocks for feature recalibration (Hu et al., 2018). ShuffleNet, GhostNet and MobileNet aim for efficiency by shuffle operations (Zhang et al., 2018), cheap feature map generation (Han et al., 2020; 2022) and depthwise separable convolutions (Howard et al., 2017). In addition to CNNs, transformers have been explored in both CV and NLP. BERT pre-trains deep bidirectional representations (Devlin et al., 2018), while (Dosovitskiy et al., 2020) and (Liu et al., 2021) apply transformers to image patches and shifted windows, respectively.

**Neural Architecture Search**    Neural Architecture Search (NAS) automates the search for optimal CNN designs, as evidenced by works such as (Dong et al., 2020; Li et al., 2022b; Baker et al., 2016; Vahdat et al., 2020; Guo et al., 2020; Chen et al., 2021; Niu et al., 2021; Guo et al., 2021). Recently, it has been extended to ViTs (Su et al., 2022). AmoebaNet evolves network blocks (Real et al., 2019), while (Zoph et al., 2018) use evolutionary algorithms to optimize cell structures. DARTS employs differentiable searching (Liu et al., 2018), and PDARTS considers architecture depths (Chen et al., 2019). (Dong & Yang, 2019) use the Gumbel-Max trick for differentiable sampling over graphs. NAS benchmarks have also been developed (Ying et al., 2019; Dong & Yang, 2020). This work

aims at the neural architecture retrieval of all existing and future neural architectures instead of those within a pre-defined search space.

**Graph Pre-training Strategy** Graph neural networks become an effective method for graph representation learning (Hamilton et al., 2017; Li et al., 2015; Kipf & Welling, 2016). To achieve generalizable and accurate representation learning on graph, self-supervised learning and pre-training strategies have been widely studied (Hu et al., 2019; You et al., 2020; Velickovic et al., 2019). Velickovic et al. (2019) used mutual information maximization for node learning. Hamilton et al. (2017) focused on edge prediction. Hu et al. (2019) employed multiple pre-training tasks at both node and graph levels. You et al. (2020) use data augmentations for contrastive learning. Different from previous works which focused on graph or node pre-training, this work pay more attention to the motifs and macro graph of neural architectures in pre-train tasks designing due to the characteristic of neural architectures.

## 3 METHODOLOGY

### 3.1 PROBLEM FORMULATION

Given a query of neural architecture $A_q$, our proposed neural architecture retrieval algorithm returns a set of neural architectures $\{A_k\}_{k=1}^K \in \mathcal{A}$, which have similar architecture as $A_q$. In order to achieve efficient searching of similar neural architectures, we propose to utilize a network $\mathcal{F}$ to map each neural architecture $A_q \in \mathcal{A}$ to an embedding $H_q$ where the embedding of similar neural architectures are clustered. We denote the set of embedding of existing neural architectures as $\mathcal{H}$. Through the derivation of embedding $H_q$ of the query neural architecture $A_q$, a set of similar neural architectures $\{A_k\}_{k=1}^K = \{A_1, A_2, ..., A_K\}$ can be found through similarity measurement:

$$H_q \leftarrow \mathcal{F}(A_q); \quad \{I_k\}_{k=1}^K \leftarrow \underset{H_i \in \mathcal{H}}{\operatorname{argsort}} \left[ \frac{H_q \cdot H_i}{\|H_q\| \cdot \|H_i\|}, K \right], \tag{1}$$

where $\operatorname{argsort}[\cdot, K]$ denotes the function to find the indices $I_k$ of the maximum $K$ values given a pre-defined similarity measurement, and we use cosine similarity in Eq. 1. With the top-$k$ similarity indices, we can acquire the $\{A_k\}_{k=1}^K$ from the candidates $\mathcal{A}$.

A successful searching of similar neural architectures requires an accurate, effective, and efficient embedding network $\mathcal{F}$. Specifically, the network $\mathcal{F}$ is expected to capture the architecture similarity and be generalized to Out-of-Distribution (OOD) neural architectures. To design network $F$, we first consider the data structure of neural architectures. Given the model definition, the computational graph can be derived from an initialized model. With the computational graph, the neural architectures can be represented by a directed acyclic graph where each node denotes the operation and edge denotes the connectivity. It is natural to apply GNNs to handle this graph-based data.

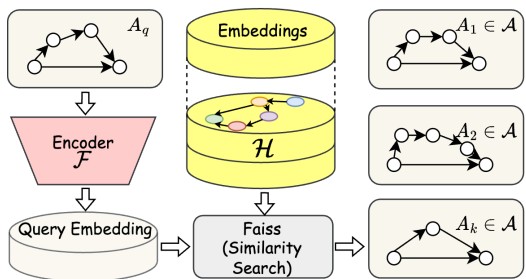

Figure 1: The definition of *Neural Architecture Retrieval* (**NAR**). This paper explores pre-training an encoder $\mathcal{F}$ to build a neural network embedding database $\mathcal{H}$ based on the architecture designs.

However, there exists some risks when it comes to neural architectures. First, the sizes of the neural architecture graphs vary significantly from one to another, and the sizes of models with state-of-the-art performance keep expanding. For example, a small number of operations are involved in AlexNet whose computational graph is in small size Krizhevsky et al. (2012), while recent vision transformer models contain massive operations and their computational graphs grow rapidly Dosovitskiy et al. (2020). Thus, given extremely large computational graphs, there exists an increasing computational burden of encoding neural architectures and it could be difficult for GNNs to capture valid architecture representations. Second, different from traditional graph-based data, there exist motifs in neural architectures. For example, ResNets contains the block design with residual connections and vision transformers contain self-attention modules He et al. (2016); Dosovitskiy et al. (2020), which are stacked for multiple times in their models. Since these motifs reflect the architecture designs, taking motifs into consideration becomes an essential step in neural architecture embedding.

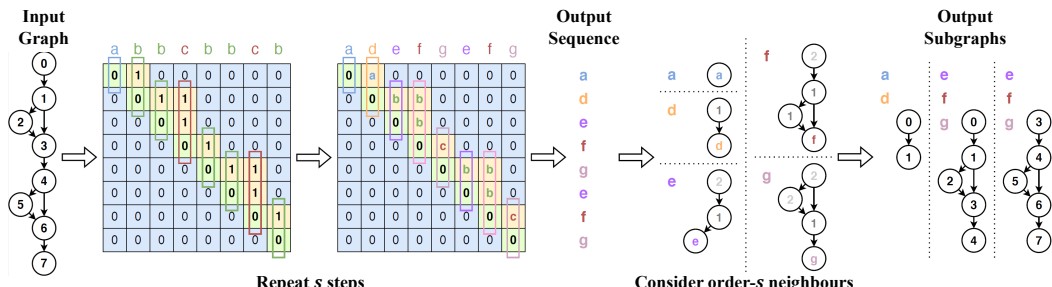

Figure 2: An illustration of motifs sampling strategy. The graph nodes encode their neighbours in adjacent matrix via an iterative manner to form the encoded node sequence where each node denotes a subgraph and motifs denote the subsequence.

## 3.2 MOTIFS IN NEURAL ARCHITECTURE

To capture the repeated designs in the neural architectures, we propose to discover the motifs in computational graphs $G$. The complexity of searching motifs grows exponentially since the size and pattern of motifs in neural architectures are not fixed. For efficient motifs mining, we introduce a new motifs sampling strategy which encodes the neighbours for each node to expand the receptive field in the graph. An illustration is shown in Figure 2. Specifically, given the computational graph $G$ with $m$ nodes, we first compute the adjacent matrix $\mathcal{M} \in \mathbb{R}^{m \times m}$ and label the neighbour pattern for each node via checking the columns of adjacent matrix. As shown in the left part of Figure 2, a new label is assigned to each new pattern of sequence in $\mathcal{M}_{1:m,i}$, where we denote the label of node $N_i$ as $C_i$. With $C$ encoding the first order neighbours, each node can be represented by a motif. Through performing this procedure by $s$ steps in an iterative manner, the receptive field can be further expanded. Formally, the node encoding can be formulated as

$$\mathcal{M}_{1:m,i}^k = \text{EN}_{\mathcal{M}}(\mathcal{M}_{1:m,i}^{k-1}), \; C_{1:m}^k = \text{EN}_C(\mathcal{M}^k), \quad \text{EN}_{\mathcal{M}}(\mathcal{M}_{j,i}^k) = \begin{cases} C_j^{k-1}, & \text{if } \mathcal{M}_{j,i}^{k-1} \neq 0 \\ \mathcal{M}_{j,i}^{k-1}, & \text{otherwise} \end{cases},$$
(2)

where $k$ denotes the encoding step, and $\text{EN}_C$ denotes the label function which assigns new or existing labels to the corresponding sequence in $\mathcal{M}^k$. With Eq. 2 after $s$ steps, the computational graph is converted to a sequence of encoded nodes $C^s$ and each node encodes order-$s$ neighbours in the graph. The motifs can be easily found in $C^s$ through discovering the repeated subsequences. In Figure 2, we illustrate with a toy example which only considers the topology in adjacent matrix without node labels and takes the parents as neighbours. However, we can easily generalize it to the scenario where both parents and children are taken into consideration as well as node labels through the modification of adjacent matrix $\mathcal{M}^1$ at the first step.

## 3.3 MOTIFS TO MACRO GRAPH

With the motifs in neural architectures, the aforementioned risks including the huge computational graph size and the involvement of motifs in neural architectures can be well tackled. Specifically, we propose to represent each motif $G_s$ as a node with an embedding $H_{sg}$ and recover the computational graph $G$ to form a macro graph $G_m$ through replacing the motifs in $G$ by the motifs embedding according to the connectivity among these motifs. An illustration of macro graph setup is shown in Figure 3 (a). All the motifs are mapped from $G_s$ to the embedding $H_{sg}$ respectively through a multi-layer graph convolutional network as

$$H_{sg}^{(l+1)} = \sigma(\hat{D}^{-\frac{1}{2}} \hat{\mathcal{M}}_{sg} \hat{D}^{-\frac{1}{2}} H_{sg}^{(l)} W^{(l)}),$$
(3)

where $l$ denotes the layer, $\sigma$ denotes the activation function, $W$ denotes the weight parameters, $\hat{D}$ denotes diagonal node degree matrix of $\hat{\mathcal{M}}_{sg}$, and $\hat{\mathcal{M}}_{sg} = \mathcal{M}_{sg} + I$ where $\mathcal{M}_{sg}$ is the adjacent matrix of motif $G_s$ and $I$ is the identity matrix. Since we propose to repeat $s$ steps in Eq. 2 to cover the neighbours in the graph, the motifs have overlapped edges, such as edge $0 \to 1$ and edge $3 \to 4$ in Figure 3 (a), which can be utilized to determine the connectivity of the nodes $H_{sg}$ in the macro graph. Based on the rule that the motifs with overlapped edges are connected, we build the macro

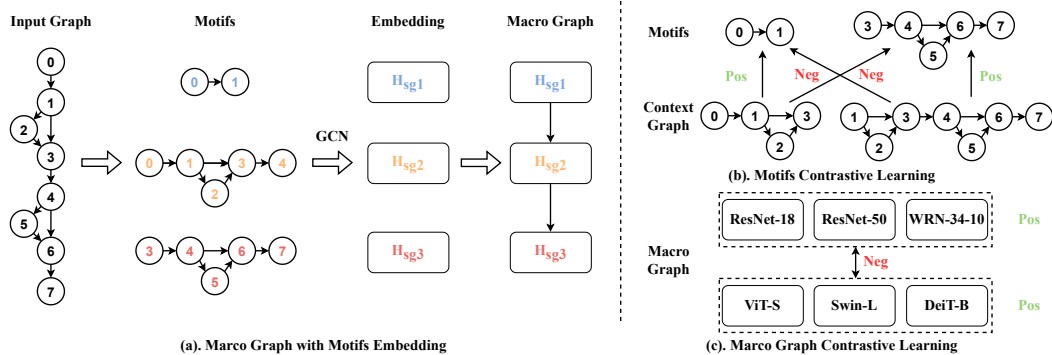

Figure 3: An illustration of macro graph setup and pre-training in motifs-level and graph-level.

graph where each node denotes the motif embedding. With macro graph, the computational burden of GCNs due to the huge graph size can be significantly reduced. Furthermore, some block and module designs in neural architectures can be well captured via motifs sampling and embedding. For a better representation learning of neural architectures, we introduce a two-stage embedding learning which involves pre-train tasks in motifs-level and graph-level respective.

### 3.4 MOTIFS-LEVEL CONTRASTIVE LEARNING

In motifs embedding, we use GCNs to obtain the motif representations. For accurate representation learning of motifs which can be better generalized to OOD motifs, we introduce the motifs-level contrastive learning through the involvement of context graph $G_c$. We define the context graph of a motifs $G_s$ as the combined graph of $G_s$ and the $k$-hop neighbours of $G_s$ in graph $G$. A toy example of context graph with 1-hop neighbours is shown in Figure 3 (b). For example, we sample the motif with nodes 0 and 1 from the graph, the context graph of this motif includes node 2 and 3 in its 1-hop neighbours. With the motifs and their context graphs, we introduce a motifs-level pre-train task in a contrastive manner. Formally, given a motif $G_s \in \mathcal{G}_s$, we denote the corresponding context graph of $G_s$ as the positive sample $G_c^+$ and the rest context graph of motifs $\mathcal{G}_s$ as the negative samples $G_c^-$. We denote the two GCN network as $\mathcal{F}_s$ and $\mathcal{F}_c$ defined in Eq. 3, the contrastive loss can be formulated as

$$\mathcal{L}_s = -\log \frac{e^{d(H_{sg}, H_c)}}{e^{d(H_{sg}, H_c)} + \sum_{k=1}^{|\mathcal{G}_s - 1|} e^{d(H_{sg}, H_k')}}, \tag{4}$$

where $H_{sg} = \mathcal{F}_s(G_s)$, $H_c = \mathcal{F}_c(G_c^+)$, $H' = \mathcal{F}_c(G_c^-)$, and $d$ denotes the similarity measurement which we use cosine distance. With Eq. 4 as the motifs-level pre-training objective, an accurate representation of motifs $H_{sg}$ can be derived in the first stage. Note that $H_c$ and $H'$ only exist in the training phase and are discarded during inference phase.

### 3.5 GRAPH-LEVEL PRE-TRAINING STRATEGY

With the optimized motifs embedding $H_{sg}^*$ from Eq. 4, we can build the macro graph $G_m$ with a significantly reduced size. Similarly, we use a GCN network $\mathcal{F}_m$ defined in Eq. 3 to embed $G_m$ as $H_m = \mathcal{F}_m(G_m)$. For the clustering of similar graphs, we propose to include contrastive learning and classification as the graph-level pre-train tasks via a low level of granularity, such as model family. An illustration is shown in Figure 3 (c). For example, ResNet-18, ResNet-50, and WideResNet-34-10 He et al. (2016); Zagoruyko & Komodakis (2016) belong to the ResNet family, while ViT-S, Swin-L, Deit-B Dosovitskiy et al. (2020); Liu et al. (2021); Touvron et al. (2021) belong to the ViT family. Formally, given a macro graph $G_m$, we denote a set of macro graphs which belong to the same model family of $G_m$ as the positive samples $\mathcal{G}_m^+$ with size $K^+$ and those not as negative samples $\mathcal{G}_m^-$ with size $K^-$. The graph-level contrastive loss can be formulated as

$$\mathcal{L}_m = -\log \frac{\sum_{k=1}^{K^+} e^{d(H_m, H_m^+(k))}}{\sum_{k=1}^{K^+} e^{d(H_m, H_m^+(k))} + \sum_{k=1}^{K^-} e^{d(H_m, H_m^-(k))}}, \tag{5}$$

where $H_m^+(k) = \mathcal{F}_m(\mathcal{G}_m^+(k))$, $H_m^-(k) = \mathcal{F}_m(\mathcal{G}_m^-(k))$. Besides the contrastive learning in Eq. 5, we also include the macro graph classification as another pre-train task which utilizes model family

as the label. The graph-level pre-training objective can be formulated as

$$\mathcal{L}_G = \mathcal{L}_m(\mathcal{F}_m; H_{sg}) + \mathcal{L}_{ce}(f; H_m, c), \tag{6}$$

where $L_{ce}$ denotes the cross-entropy loss, $f$ denotes the classifier head, and $c$ denote the ground-truth label. With the involvement of contrastive learning and classification in Eq. 6, a robust graph representation learning can be achieved where the embedding $H_m$ with similar neural architecture designs are clustered while those different designs are dispersed. The two-stage learning can be formulated as

$$\min_{\mathcal{F}_m, f} \mathcal{L}_G(\mathcal{F}_m, f; H_{sg}^*, c), \qquad \textbf{s.t.} \qquad \underset{\mathcal{F}_s, \mathcal{F}_c}{\operatorname{argmin}} \mathcal{L}_s(\mathcal{F}_s, \mathcal{F}_c; G). \tag{7}$$

In inference phase, the optimized GCNs of embedding macro graph $\mathcal{F}_m$ and motifs $\mathcal{F}_s$ are involved, the network $\mathcal{F}$ in Eq. 1 can be reformulated as

$$\mathcal{F}(A) = \mathcal{F}_m\Big(\text{Agg}[\mathcal{F}_s(\text{Mss}(A))]\Big), \tag{8}$$

where Agg denotes the aggregation function which aggregates the motifs embedding to form macro graph, and Mss denotes the motifs sampling strategy.

# 4 EXPERIMENTS

In this section, we conduct experiments with both real-world neural architectures and NAS architectures to evaluate our proposed subgraph splitting method and two-phase graph representation learning method. We also transfers models pre-trained with NAS architectures to real-world neural architectures.

## 4.1 DATASETS

**Data Collection**: We crawl $12,517$ real-world neural architecture designs from public repositories which have been formulated and configured. These real-world neural architectures cover most deep learning tasks, including image classification, image segmentation, object detection, fill-mask modeling, question-answering, sentence classification, sentence similarity, text summary, text classification, token classification, language translation, and automatic speech recognition.

We extract the computational graph generated by the forward propagation of each model. Each node in the graph denotes an atomic operation in the network architecture. The data structure of each model includes: the model name, the repository name, the task name, a list of graph edges, the number of FLOPs, and the number of parameters. Besides, we also build a dataset with $30,000$ NAS architectures generated by algorithms. The architectures follow the search space of DARTS Liu et al. (2018) and are split into 10 classes based on the graph editing distance.

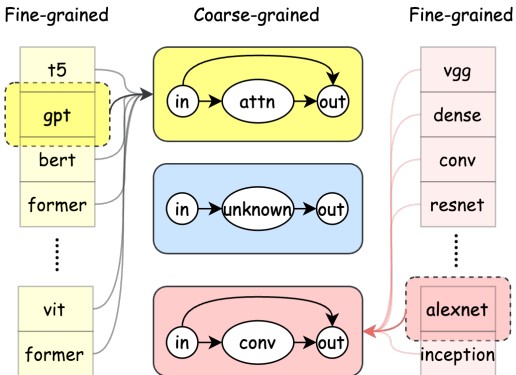

Figure 4: Coarse-grained and fine-grained classes.

**Data Pre-Processing**: We scan the key phrases and operations from the raw graph edges of the model architecture. We identify the nodes in the graph based on the operator name and label each edge as index. Each node is encoded with a one-hot embedding representation. The key hints such as 'former', 'conv' and 'roberta' extracted by regular expressions tools represent a fine-grained classification, which are treated as the ground truth label of the neural network architecture in Eq. 6. We then map the extracted fine-grained hints to the cnn-block, attention-block and other block as the coarse-grained labels. Due to the involvement of motifs in neural architectures, we extract the main repeated block cell of each model by the method presented in the 3.3. Also, we scan these real-world neural architectures and extracted 89 meaningful operators like "Addmm", "NativeLayerNorm", "AvgPool2D" and removed useless operators "Tbackward" and "AccumulateBackward". The data structure of each pre-processed record consists of model name, repository name, task name, unique operators, edge index, one-hot embeddings representation, coarse-grained label. We divided the pre-processed data records to train/test splits (0.9/0.1) stratified based on the fine-grained classes for testing the retrieval performance on the real-world neural architectures.

| Dataset | Method | MRR | | | MAP | | | NDCG | | |
|---------|--------|-----|-----|-----|-----|-----|-----|------|-----|-----|
| | | Top-20 | Top-50 | Top-100 | Top-20 | Top-50 | Top-100 | Top-20 | Top-50 | Top-100 |
| **Real** | GCN | 0.737 | 0.745 | 0.774 | 0.598 | 0.560 | 0.510 | 0.686 | 0.672 | 0.628 |
| | GAT | 0.756 | 0.776 | 0.787 | 0.542 | 0.541 | 0.538 | 0.610 | 0.598 | 0.511 |
| | Ours | **0.825** | **0.826** | **0.826** | **0.593** | **0.577** | **0.545** | **0.705** | **0.692** | **0.678** |
| **NAS** | GCN | 1.000 | 1.000 | 1.000 | 0.927 | 0.854 | 0.858 | 0.953 | 0.902 | 0.906 |
| | GAT | 1.000 | 1.000 | 1.000 | 0.941 | 0.899 | 0.901 | 0.961 | 0.933 | 0.935 |
| | Ours | **1.000** | **1.000** | **1.000** | **0.952** | **0.932** | **0.935** | **0.969** | **0.960** | **0.958** |

Table 1: Comparison with baselines on real-world neural architectures and NAS data.

| Dataset | Splitting | MRR | | | MAP | | | NDCG | | |
|---------|-----------|-----|-----|-----|-----|-----|-----|------|-----|-----|
| | | Top-20 | Top-50 | Top-100 | Top-20 | Top-50 | Top-100 | Top-20 | Top-50 | Top-100 |
| **Real** | Node Num | 0.807 | 0.809 | 0.809 | 0.551 | 0.539 | 0.537 | 0.694 | 0.682 | 0.667 |
| | Motif Num | 0.817 | 0.820 | 0.823 | 0.591 | 0.522 | 0.518 | 0.692 | 0.669 | 0.661 |
| | Random | 0.801 | 0.802 | 0.804 | 0.589 | 0.543 | 0.536 | 0.699 | 0.675 | 0.668 |
| | Ours | **0.825** | **0.826** | **0.826** | **0.593** | **0.577** | 0.545 | **0.705** | **0.692** | **0.678** |
| **NAS** | Node Num | 0.999 | 0.999 | 0.999 | 0.941 | 0.885 | 0.883 | 0.962 | 0.926 | 0.924 |
| | Motif Num | 0.998 | 0.998 | 0.998 | 0.931 | 0.872 | 0.874 | 0.956 | 0.917 | 0.919 |
| | Random | 1.000 | 1.000 | 1.000 | 0.919 | 0.826 | 0.824 | 0.949 | 0.881 | 0.883 |
| | Ours | **1.000** | **1.000** | **1.000** | **0.952** | **0.936** | **0.935** | **0.969** | **0.957** | **0.958** |

Table 2: Evaluation of different graph split methods on real-world and NAS architectures.

## 4.2 EXPERIMENTAL SETUP

In order to ensure the fairness of the implementation, we set the same hyperparameter training recipe for the baselines, and configure the same input channel, output channel, and number layers for the pre-training models. This encapsulation and modularity ensures that all differences in their retrieval performance come from a few lines of change. In the test stage, each query will get the corresponding similarity rank index and is compared with the ground truth set. We utilize the three most popular rank-aware evaluation metrics: mean reciprocal rank (MRR), mean average precision (MAP), and Normalized Discounted Cumulative Gain (NDCG) to evaluate whether the pre-trained embeddings can retrieve the correct answer in the top k returning results. We now demonstrate the use of our pre-training method as a benchmark for neural network search. We first evaluate the ranking performance of the most popular graph embedding pre-training baselines. Afterwards we investigate the performance based on the splitting subgraph methods and graph-level pre-training loss function design. Then we conduct the ablation studies on the loss functions to investigate the influence of each sub-objective and show the cluster figures based on the pre-training class.

## 4.3 BASELINES

We evaluate the ranking performance of our method by comparing with two mainstream graph embedding baselines, including Graph Convolutions Networks (GCNs) which exploit the spectral structure of graph in a convolutional manner Kipf & Welling (2016) and Graph attention networks (GAT) which utilizes masked self-attention layers Veličković et al. (2017). For each baseline model, we feed the computational graph edges as inputs. The model self-supervised learning by contrastive learning and classification on the mapped coarse-grained label. Each query on the test set gets a returned similar models list and the performance is evaluated by comparing the top-k candidate models and ground truth of similarity architectures. Table 1 lists the rank-aware retrieval scores on the test set. We observed that our pre-training method outperforms baselines by achieving different degrees of improvement. On the dataset, the upper group of Table 1 demonstrates the our pre-training method outperforms the mainstream popular graph embedding methods. The average score of MRR, MAP and NDCG respectively increased by $+5.4\%$, $+2.9\%$, $+13\%$ on the real-world neural architectures search. On the larger nas datasets, our model also achieved considerable enhancement of the ranking predicted score with $+1.1\%$, $+3.4\%$ on map@20, map@100 and with $+8\%$, $+2.9\%$ on NDGC@20 and NDCG@100.

| Dataset | Objective | MRR | | | MAP | | | NDCG | | |
|---|---|---|---|---|---|---|---|---|---|---|
| | | Top-20 | Top-50 | Top-100 | Top-20 | Top-50 | Top-100 | Top-20 | Top-50 | Top-100 |
| **Real** | CE | 0.824 | 0.828 | 0.829 | 0.583 | 0.573 | 0.539 | 0.703 | 0.693 | 0.692 |
| | CL | 0.565 | 0.572 | 0.573 | 0.334 | 0.348 | 0.373 | 0.502 | 0.455 | 0.451 |
| | CE+CL | **0.825** | **0.826** | **0.826** | **0.593** | **0.577** | 0.545 | **0.705** | **0.692** | **0.678** |
| **NAS** | CE | 1.000 | 1.000 | 1.000 | **0.953** | 0.921 | **0.923** | 0.969 | 0.952 | 0.950 |
| | CL | 0.925 | 0.925 | 0.925 | 0.750 | 0.658 | 0.656 | 0.829 | 0.762 | 0.764 |
| | CE+CL | **1.000** | **1.000** | **1.000** | 0.952 | **0.932** | 0.935 | **0.969** | **0.955** | **0.958** |

Table 3: Ablation study of different loss terms (CE: Cross Entropy; CL: Contrastive Learning).

## 4.4 SUBGRAPH SPLITTING

We compare our method for splitting subgraphs with three baselines. Firstly, we use two methods to uniformly split subgraphs, where the number of nodes in each subgraph (by node number) or the number of subgraphs (by motif number) are specified. If the number of nodes in each subgraph is specified, architectures are split into motifs of the same size. Consequently, large networks are split into more motifs, while small ones are split into fewer motifs. If the number of subgraphs is specified, different architectures are split into various sizes of motifs to ensure the total number of motifs is the same. Then, we also use a method to randomly split subgraphs, where the sizes of motifs are limited to a given range. We report the results in Table 2. As can be seen, our method can consistently outperform the baseline methods on both real-network and NAS architectures. When comparing the baseline methods, we find that for NAS architectures, splitting by node number and by motif number reaches similar performances. It might be because NAS architectures have similar sizes. For real-network architectures, whose sizes vary, random splitting reaches the best NDCG among all baselines, and splitting by motif number reaches the best MRR. Considering MAP, splitting by motif number achieves the best Top-20 performance, but splitting by node number achieves the best Top-100 performance. On the other hand, our method can consistently outperform the baselines. This phenomenon implies that the baselines are not stable under different metrics when the difference in the size of architectures is non-negligible.

## 4.5 OBJECTIVE FUNCTION

Since our graph-level pre-training is a multi-objectives task, it is necessary to explore the effectiveness of each loss term by removing one of the components. All hyperparameters of the models are tuned using the same training receipt as in Table 5. Table 3 provides the experimental records of different loss terms. In terms of the Real dataset, the model trained with both CE and CL (CE+CL) outperforms the models trained with either CE or CL alone across almost all metrics. Specifically, the MRR scores for CE+CL are 0.825, 0.826, and 0.826 for Top-20, Top-50, and Top-100, respectively. These scores are marginally better than the CE-only model, which has MRR scores of 0.824, 0.828, and 0.829, and significantly better than the CL-only model, which lags with scores of 0.565, 0.572, and 0.573. Similar trends are observed in MAP and NDCG scores, reinforcing the notion that the combined loss term is more effective. For the NAS dataset, the CE+CL model again demonstrates superior performance, achieving perfect MRR scores of 1.000 across all rankings. While the CE-only model also achieves perfect MRR scores, it falls short in MAP and NDCG metrics, especially when compared to the combined loss term. The ablation study reveals that a multi-objective approach involving both graph-level contrastive learning and coarse label classification is most effective in enhancing neural architecture retrieval performance. Furthermore, the contrastive loss term, while less effective on its own, plays a crucial role in boosting performance when combined with cross-entropy loss.

## 4.6 TRANSFER LEARNING

We also monitor whether NAS pre-training benefits the structure similarity prediction of the real-world network. For this, we design the experiment of transferring the pre-trained model from the NAS datasets to initialize the model for pre-training on the real-world neural architectures. The results demonstrated in Table 4 shows the model pre-trained on the real-world neural architectures achieves an improvement on most evaluation metrics, which reveals the embeddings pre-trained by initialized model obtains the prior knowledge and get benefits from the NAS network architecture

| Training Method | MRR | | | MAP | | | NDCG | | |
|---|---|---|---|---|---|---|---|---|---|
| | Top-20 | Top-50 | Top-100 | Top-20 | Top-50 | Top-100 | Top-20 | Top-50 | Top-100 |
| Training from Scratch | **0.825** | 0.826 | 0.826 | 0.593 | 0.577 | 0.545 | 0.705 | 0.692 | 0.678 |
| Pre-training with NAS | 0.821 | **0.838** | **0.839** | **0.596** | **0.584** | **0.573** | **0.712** | **0.703** | **0.706** |

Table 4: Transfer model pre-trained with NAS architectures to real-world neural architectures.

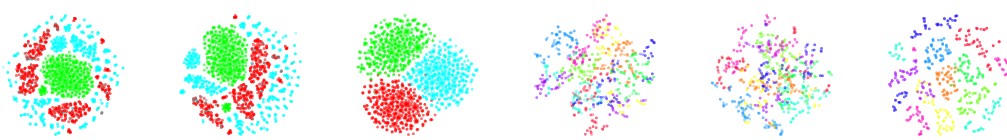

(a) GCN on real   (b) GAT on real   (c) Ours on real   (d) GCN on NAS   (e) GAT on NAS   (f) Ours on NAS

Figure 5: Visualization of learnt embeddings. The number of dimensions is reduced by t-SNE. (a, b and c are embeddings of real-world neural architectures, and d, e and f are embeddings of NAS architectures)

searching. And with the increment of top-k of rank lists, the model unitized with NAS pre-training yields a higher score compared with the base case, which means enlarging the search space could boost the similarity model structures by using the model that transferred from NAS.

## 4.7 VISUALIZATION

Besides the quantitative results provided in Table 1, we further provide qualitative results through the visualization of cluttering performance in Figure 5. To illustrate the superiority of our method over other baselines, we include both GCN and GAT for comparison. For visualization, we apply t-SNE Van der Maaten & Hinton (2008) to visualize the high-dimensional graph embedding through dimensionality reduction techniques. As shown in Figure 5 (a), (b), and (c), we visualize the clustering performance of real-world neural architectures on three different categories, including attention-based blocks (green), CNN-based blocks (red), ang other blocks (blue). Comparing the visualization results on real-world neural architectures, it is obvious that both GCN and GAT cannot perform effective clustering of the neural architectures with the blocks from same category. On the contrary, our proposed method can achieve better clustering performance than other baselines. Similarly, we conduct visualization on the NAS data. We first sample ten diverse neural architectures from the entire NAS space as the center points of ten clusters respectively. Then we evaluate the clustering performance of neural architectures sampled around center points that have similar graph editing distance from these sampled center points. The results are shown in Fig. 5 (d), (e), and (f). Consistent with the results on real-world data, we can see that our method can achieve better clustering performance on NAS data with clear clusters and margins, which provides strong evidence that our method can achieve accurate graph embedding for neural architectures.

## 5 CONCLUSION

In this paper, we define a new and challenging problem Neural Architecture Retrieval which aims at recording valuable neural architecture designs as well as achieving efficient and accurate retrieval. Given the limitations of existing GNN-based embedding techniques on learning neural architecture representations, we introduce a novel graph representation learning framework that takes into consideration the motifs of neural architectures with designed pre-training tasks. Through sufficient evaluation with both real-world neural architectures and NAS architectures, we show the superiority of our method over other baselines. Given this success, we build a new dataset with 12k different collected architectures with their embedding for neural architecture retrieval, which benefits the community of neural architecture designs.

ACKNOWLEDGMENTS

This work was supported in part by the Australian Research Council under Projects DP240101848 and FT230100549.

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

# A APPENDIX

## A.1 APPLICATION DEMOS

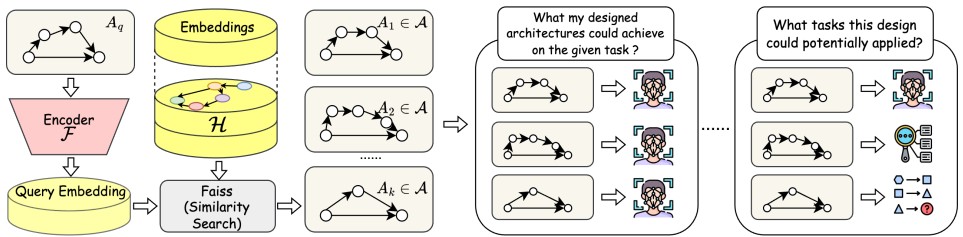

Figure 6: A depiction of the potential downstream applications for NAR.

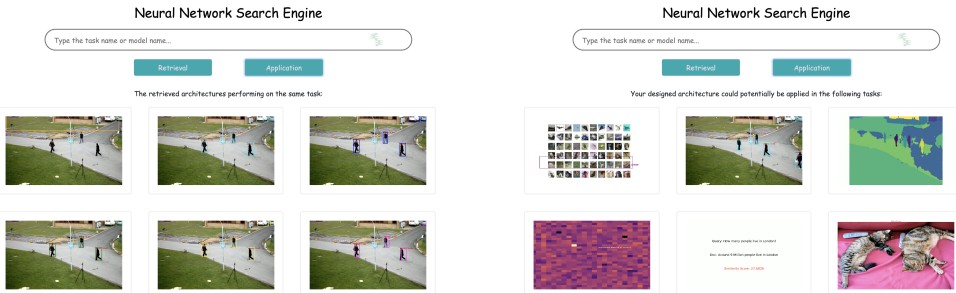

(a) Architectures Retrieval for the specific task.      (b) Architectures Retrieval for various applications.

Figure 7: Cases of potential downstream tasks based on the NAR.

## A.2 ALGORITHM

---
**Algorithm 1** NAR Pre-training
---
**Require:** A set $\mathcal{G}$ of computational graph
**Ensure:** $\mathcal{F}_s^*$ and $\mathcal{F}_m^*$
  1: **for** $G \in \mathcal{G}$ **do**                 ▷ motifs-level CL, Fig. 3
  2:      Get $\mathcal{G}_s$ from $G$               ▷ motifs sampling, Fig. 2
  3:      **for** $G_s \in \mathcal{G}_s$ **do**
  4:          Get $G_c^+, G_c^-$ based on $G_s$
  5:          Calculate $\mathcal{L}_s$ with Eq. 4 and update $\mathcal{F}_s, \mathcal{F}_c$
  6:      **end for**
  7: **end for**
  8: **for** $G \in \mathcal{G}$ **do**                 ▷ graph-level CL, Fig. 3 (c)
  9:      Get $\mathcal{G}_s$ from G               ▷ motifs sampling, Fig. 2
10:      Build $G_m$ with $\mathcal{F}_s^*$         ▷ build macro graph, Fig. 3 (a)
11:      Calculate $\mathcal{L}_G$ with Eq. 6 and update $\mathcal{F}_m$
12: **end for**
---

## A.3 DETAILS

## A.4 NEURAL ARCHITECTURE GENERATION

To generate diverse neural architectures, we follow the search space design for neural architecture search (NAS) in DARTS Liu et al. (2018), which considers neural architectures as directed acyclic graphs (DAGs). A difference is that DARTS treats operations as edge attributes, while we insert an additional node representing an operation to each edge with an operation for consistency. Our space

| | EPs | BS | Layers | LR | Emb | Drop |
|---|---|---|---|---|---|---|
| Motifs CL | 5 | 256 | 3 | 1e-2 | 512 | - |
| Graph CL | 15 | 512 | 3 | 1e-3 | 512 | 0.1 |
| Baselines | 15 | 512 | 3 | 1e-3 | 512 | 0.1 |

Table 5: Pre-training Recipes. EPs: Epochs; BS: Batch size; LR: Learning rate.

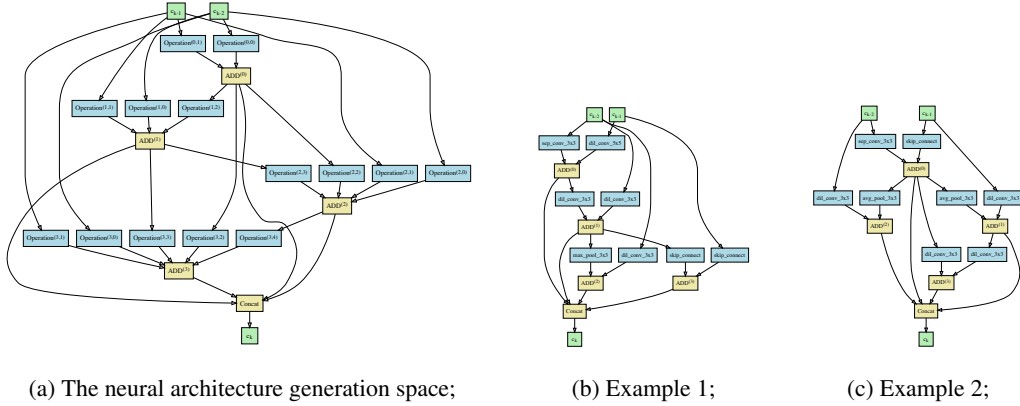

(a) The neural architecture generation space;  (b) Example 1;  (c) Example 2;

Figure 8: The neural architecture generation space and two examples.

for architecture generation is as shown in Fig. 8 (a). We also provide two examples in Fig. 8 (b) and (c).

A neural architecture is used to build a cell, and cells are repeated to form a neural network. Each cell $c_k$ with $k = 0, \ldots, K$ takes inputs from two previous cells $c_{k-2}$ and $c_{k-1}$. The beginning of a network is a stem layer with an convolutional layer. The first cell $c_0$ is connected to the stem layer, and the second cell $c_1$ connected to both $c_0$ and the stem layer. In the other word, $c_{-2}$ and $c_{-1}$ both refer to the stem layer. Finally, the last cell $c_K$ is connected to a global average pooling and a fully connected layer to generate the network output.

In each cell, there are 4 "ADD" nodes $\text{ADD}^{(i)}$ with $i = 0, 1, 2, 3$. The node $\text{ADD}^{(i)}$ can be connected to $c_{k-2}$, $c_{k-1}$, or $\text{ADD}^{(j)}$ with $0 \le j < i$. In practice, the number of connections is limited to 2. For each connection, we insert a node to represent an operation. Operations are chosen from 7 candidates, including **skip connection** (skip_connect), $\mathbf{3 \times 3}$ **max pooling** (max_pool_3x3), $\mathbf{3 \times 3}$ **average pooling** (avg_pool_3x3), $\mathbf{3 \times 3}$ or $\mathbf{5 \times 5}$ **separable convolution** (sep_conv_3x3 or sep_conv_5x5), and $\mathbf{3 \times 3}$ or $\mathbf{5 \times 5}$ **dilated convolution** (dil_conv_3x3 or dil_conv_5x5). Finally, a "Concat" node is used to concatenate the 4 "ADD" nodes as the cell output $c_k$.

