# OpenReview forum: "Neural Architecture Retrieval"
_ICLR.cc/2024/Conference — ICLR 2024 poster_

### Official Review · Reviewer_BEfk · 2023-10-28

**Soundness:** 4 excellent
**Presentation:** 3 good
**Contribution:** 4 excellent
**Rating:** 8
**Confidence:** 5

**Summary:**

This is a very interesting paper, considering the burgeoning field of neural architecture designs which has made it increasingly challenging for researchers to position their contributions or establish relationships between their designs and existing ones. The paper presents a novel problem of neural architecture retrieval, aiming to automate the discovery of similar neural architectures. The authors identify the limitations of existing graph pre-training strategies, which are unable to handle the computational graphs in neural architectures due to the graph size and motifs. They propose a creative solution by dividing the graph into motifs, which are then used to reconstruct the macro graph, addressing the identified issues. Moreover, the introduction of multi-level contrastive learning is put forth to attain precise graph representation learning. The paper boasts extensive evaluations of both human-designed and synthesized neural architectures, showcasing the superiority of their proposed algorithm. The authors also construct a valuable dataset comprising 12k real-world network architectures along with their embeddings, laying a solid foundation for neural architecture retrieval. This endeavor may address a pressing issue in the domain with pre-training an embedding database for finding and comparing neural architectures. The paper potentially opens up a new domain within the neural architecture community, paving the way for more organized and efficient exploration in this field by building an embedding database, and the video demo shows promising applications.

**Strengths:**

* The motivation behind the paper is well-articulated and resonates with the ongoing challenges faced by researchers in situating their contributions amidst a plethora of existing neural architectures. The introduction of Neural Architecture Retrieval as a solution to automate the discovery of similar neural architectures is timely and could significantly alleviate the existing bottleneck.
* The methods used in the paper sound and well-justified. The logic behind each step of the solution is sound and reasonable, showcasing a thorough understanding of the challenges at hand. For instance, the approach to addressing the repeated design of blocks is practical and applicable, demonstrating a commendable level of methodological rigor and a pragmatic stance.
* The evaluation metrics, particularly the information retrieval scores, are promising, especially when applied to the nas dataset. This suggests that the proposed methods are effective and could potentially set a new standard in evaluating neural architectures.
* Contribute a 12k real-world computational graph and its corresponding embedding database. The availability of such a dataset could spur further research and development in the domain of neural architecture retrieval and related areas.

**Weaknesses:**

* Page 5 Section 3.4 Eq 4: The objective of the first stage may have a better way. The encoder encodes the architecture into motifs, and then concatenates the embeddings.  Directly sampling the highest-frequency motifs $H$ to represent the main design for large models may more reasonable, especially considering that the concatenation of motif embeddings cannot backpropagate the gradients without two stages.

* Page 7 Section 4.3: The details of the evaluation part are lacking. Although it covers the mainstream information retrieval scores, the parameters of the computational score are still unclear. For example, when testing the NDCG, is graded relevance or non-graded relevance used? It would be better to provide a formula here.

**Questions:**

Page 5 Section 3.4 Eq 4: What role does the context graph $G_s$ play?

Page 6, Section 4.1: How were the real-world repositories collected, and how were the models obtained? Will the 12k real-world computational graphs, along with corresponding labels, be made available for follow-up work?

---

> ### Author Response · Authors · 2023-11-15
> **Response to Reviewer BEfk**
>
> **Q1** "Page 5 Section 3.4 Eq 4: ... without two stages."
>
> **A2** While sampling the highest-frequency motifs might seem reasonable in some contexts, this approach has limitations when dealing with architectures that share similar motifs. Relying solely on the most frequent motifs could lead to a lack of diversity in the learned representations, making it challenging for the model to discern the nuanced differences between various architectures. Our methodology, as described in the paper, addresses this potential issue. By considering all sampled motifs in the first stage, we ensure a comprehensive understanding of the architectural diversity. The subsequent concatenation of these motifs into macro graph embeddings, post motifs pre-training, allows for a more holistic and detailed representation. This approach not only captures the essence of the most frequent motifs but also incorporates the subtleties of less common yet potentially significant ones. This strategy ensures a richer and more effective learning process, as opposed to a singular focus on high-frequency motifs.
>
> **Q2** "Page 7 Section 4.3: The details of the evaluation part are lacking. Although it covers the mainstream information retrieval scores, the parameters of the computational score are still unclear. For example, when testing the NDCG, is graded relevance or non-graded relevance used? It would be better to provide a formula here."
>
> **A2** We reference these metrics due to their widespread application, and we employ non-graded relevance, which is prevalent in information retrieval tasks. Further details on the evaluation metrics are provided as follows: For $i$-th query:
>
> - **MRR** = $\frac{1}{|Q|}\sum_{i=1}^{|Q|}\frac{1}{r_i}$, where $|Q|$ is the total number of queries, $r_i$ is the rank of the first relevant architecture;
> - **MAP** = $\frac{1}{|Q|}\sum_{i=1}^{|Q|}\frac{1}{m_i}\sum_{k=1}^{m_i}P(k)$ where $m_i$ is the number of relevant architectures, and $P(k)$ is the precision at cut-off $k$ in the ranked sequence of architectures;
> - **NDCG** = $ \frac{1}{|Q|} \sum_{i=1}^{|Q|} \frac{1}{Z_i} \sum_{k=1}^{n_i} \frac{2^{R_i^k} - 1}{\log_2(k+1)} $ where $R^{i}_{k}$ is the relevance of the result at position $k$, $n_i$ is the number of retrieved architectures and $Z_i$ is a normalization factor.
>
>
>
>
> **Q3** "Page 5 Section 3.4 Eq 4: What role does the context graph play?"
>
> The roles of the context graph $ G_c $ in the paper:
>
> - The context graph $ G_c $ is crucial for accurately representing motifs, particularly for generalizing to out-of-distribution (OOD) motifs. It extends the representation beyond individual motifs to include their contextual information.
>
> - $ G_c $ plays a key role in the contrastive learning setup. The context graph of a motif $ G_s $ is considered a positive sample $ G^{+}_c $, and the context graphs of other motifs serve as negative samples $ G^{-}_c $. This setup is vital for learning distinct features of motifs.
>
>
> - The context graph is used in the motifs-level pre-training objective. The contrastive loss function, which utilizes the embeddings from the GCN networks $ F_s $ and $ F_c $, is designed to effectively distinguish between $ G^{+}_c $ and $ G^{-}_c $, thereby improving the motif representation learning process.
>
> **Q4** "Page 6, Section 4.1: How were the real-world repositories collected, and how were the models obtained? Will the
> 12k real-world computational graphs, along with corresponding labels, be made available for follow-up work?"
>
> We collect datasets from the Huggingface Hub and PyTorch Hub. To obtain the computational graphs, we forward the model and trace every operator in the neural network. Yes, the dataset will be made public for use.

---

### Official Review · Reviewer_wPum · 2023-10-31

**Soundness:** 3 good
**Presentation:** 3 good
**Contribution:** 3 good
**Rating:** 8
**Confidence:** 5

**Summary:**

The paper presents a new problem of Neural Architecture Retrieval, which aims to find similar neural architectures from a large collection of existing and potential ones. The paper introduces a novel graph representation learning framework that leverages the motifs of neural architectures and contrastive learning to achieve accurate and efficient retrieval. The authors also construct a new dataset of 12k real-world neural architectures with their embeddings and evaluates the proposed method on both real-world and NAS architectures, showing its superiority over baselines. However, the paper has some issues that need to be addressed. My detailed comments are as follows.

**Strengths:**

1.	The paper presents a new problem called Neural Architecture Retrieval (NAR). This problem is to find similar neural architectures quickly and easily from a big pool of existing and possible designs. It’s a smart way to make the search for neural architectures simpler and more efficient.

2.	The authors propose a novel and easy-to-understand graph representation learning framework that addresses the computational graph in neural architectures. This framework adopts motifs of neural architectures and multi-level contrastive learning for accurate graph representation learning.

3.	The paper contributes the community by creating a new dataset with 12k real-world network architectures and their embeddings. This dataset is specifically designed for neural architecture retrieval and demonstrates the effectiveness of the retrieval algorithm. It’s a helpful benchmark for everyone working in this field.

4.	The experimental results on both human-designed and synthesized neural architectures benchmarks verify the effectiveness of the proposed method.

**Weaknesses:**

1.	The proposed NAR appears to focus solely on the topological similarity of architectures. However, it is important to note that the similarity between architectures can vary across different tasks or datasets. For instance, certain architectures might yield comparable results in image classification but diverge significantly in performance when applied to other tasks. Could the authors provide additional insights and elaborations on this matter?

2.	The authors introduce a motif-level contrastive learning approach, wherein the corresponding context graph is treated as the positive sample. However, this raises a concern as there may be other context graphs that also encompass the same motifs, yet they are deemed as negative samples, which seems unreasonable. Could the authors provide further clarification and justification for this aspect of their methodology?

3.	It is unclear how the authors collect/annotate the ground-truth label of the architecture dataset. Without such labels, it is infeasible to calculate the correlation between the different networks. Please provide more details.

4.	In the "Related Work" section, it would enhance the manuscript's thoroughness if the authors could offer a more comprehensive discussion on mainstream NAS methods. This should include an exploration of reinforcement learning-based NAS methods, as referenced in [A-G].

[A] Designing Neural Network Architectures using Reinforcement Learning. ICLR 2017.

[B] Learning Transferable Architectures for Scalable Image Recognition. CVPR 2018.

[C] UNAS: Differentiable Architecture Search Meets Reinforcement Learning. CVPR 2020.

[D] Breaking the Curse of Space Explosion: Towards Efficient NAS with Curriculum Search. ICML 2020.

[E] Contrastive Neural Architecture Search with Neural Architecture Comparators.  CVPR 2021.

[F] Disturbance-immune Weight Sharing for Neural Architecture Search. Neural Networks 2021.

[G] Towards Accurate and Compact Architectures via Neural Architecture Transformer. TPAMI 2021.

**Questions:**

1. In Page 2 line 8, “exact” should be “exactly”.

2. In Page 2 line 11, “through” should be “by”.

---

> ### Author Response · Authors · 2023-11-15
> **Response to Reviewer wPum**
>
> **Q1**: "The proposed NAR appears to focus...Could the authors provide additional insights and elaborations on this matter?"
>
>
> **A1**  We are genuinely grateful for your observation and find the reviewer's question to be extremely insightful. We agree that there is an inherent relationship among architectures, tasks, and datasets, and acknowledge that some architectures perform differently in various tasks. This paper provides an important first step in gathering architecture designs to build a database for retrieving similar architectures, laying the groundwork for further studies on their impactful insights. A general research study into their inner connections may be beyond the scope of this paper. However, we propose a potential solution: observing their overlap scores in clustered embeddings from different perspectives (architectures, tasks).
> For instance, we can first retrieval a set of neural architectures embeddings doing object detection and compute the center point of these embeddings as the approximated neural architecture embedding for object detection task. Now we derive a set of similar ResNet variants with their embeddings through neural architecture retrieval. Through computing the similarity between ResNet variants embeddings and the approximated neural architecture embedding for object detection task, we can find the ResNet variant that is suitable for object detection task.
>
> **Q2** "The authors introduce ... Could the authors provide further clarification and justification for this aspect of their methodology?"
>
> **A2** In our approach, we initially identify frequent subgraphs as motifs, enabling us to recognize identical motifs. When choosing negative context graphs, we extract them from context graphs with motifs at their centers that differ from the current one. Since the center motifs are different, the negative context graph cannot include the same motif. In this way, we can avoid this problem.
>
> **Q3** It is unclear ... Please provide more details.
>
> **A3** We are grateful for your observation. We express our sincere gratitude for your detailed and insightful review. To clarify, the ground truth comprises the fine-grained classes demonstrated in Fig 4 (left and right bars). We describe the ground truth on page 6 in the 'Data Processing' section,  as follows: 'The key hints... are treated as the ground truth label of the neural network architecture.
>
> **Q4** "In the "Related Work" section...NAS methods, as referenced in [A-G]."
>
> **A4**
> We deeply appreciate your provide literatures.
>
> We will cite the insightful literatures in the NAS section and here is the discussion:
>
> In RL-based NAS, the search for an optimal neural network architecture is framed as a sequential decision-making process, where an RL agent learns to make decisions (e.g., choosing layers, connections, or hyperparameters) to maximize a reward signal, typically linked to the performance of the generated architectures on a validation set. This approach allows for automated and efficient discovery of high-performing neural network architectures tailored to specific tasks or datasets. However, the focus of the NAR problem is to directly retrieve architectures from an existing set. Integrating insights from both RL-based NAS and NAR could yield promising ideas, but such a synthesis might extend beyond the scope of this paper.
>
>
> **Q5** ​"In Page 2 line 8, “exact” should be “exactly”. In Page 2 line 11, “through” should be “by”."
>
> **A5**
> Heartfelt thanks for bringing this to our attention. We will correct in the revised version.

---

### Official Review · Reviewer_NBZG · 2023-11-01

**Soundness:** 3 good
**Presentation:** 3 good
**Contribution:** 2 fair
**Rating:** 6
**Confidence:** 5

**Summary:**

This paper defines a new question called Neural Architecture Retrieval (NAR) which returns a set of similar neural architectures given a query neural architecture. To address NAR problem, this paper proposes to split the graph into several motifs and rebuild the graph through treating motifs as nodes in a macro graph. Then this paper uses two-level pretrain task to train the architecture representation for retrieval.

**Strengths:**

1. This work proposes a large new NAS dataset of 12k real-world neural architectures rather than pre-define search space. In my opinion, this thing makes a lot of sense for the NAS field which can explore the diversity of search space species architectures.

2. The idea uses motifs to encode architecture and reduce the graph size is novel and reasonable to encode architecture and capture the connection between structures in one architecture.

3. The two-level pretrain task to train the architecture's embedding is reasonable and effective.

**Weaknesses:**

1. In my opinion, the NAR problem to find similar architectures for the query architecture doesn't seem particularly significant for real-world usage, Can the author point out the need for this problem and more application scenarios?

2. Some retrieval-based papers[1-3] for giving a query dataset to search architectures should be discussed.

3. Another question is that I want to know how much performance can be achieved by retrieving similar models only based on the architecture corresponding to the text description or the code by a language mode like Chatgpt or other language-based retrieval model.

Refs:

[1] Task-Adaptive Neural Network Search with Meta-Contrastive Learning. NeurIPS 2021.

[2] MetaGL: Evaluation-Free Selection of Graph Learning Models via Meta-Learning. ICLR 2023

[3] Retrieving GNN Architecture for Collaborative Filtering. CIKM 2023

**Questions:**

One small question is that I can't find how the ground truth topk set is constructed, can the author describe it in detail?

---

> ### Author Response · Authors · 2023-11-15
> **Response to Reviewer NBZG**
>
> **Q1** "...more application scenarios?"
>
> **A1** Heartfelt thanks for bringing this to our attention; the reviewer's question demonstrates great insight. Here are some applications scenarios:
>
>
> - **Architecture Hub for Diversity & Inclusion.** NAR collect architectures from different source, help professionals locate and engage with existing diversity-focused architectures support for corrective project, to usage inspiration on reference architectures. When scaling up to millions of architectures, it still could quickly retrieve architectures accurately. It could serve as a search engine for finding related network designs, research studies, and current practices based on architecture designs.
>
> - **Architecture Application Discovery.**
> In a typical scenario, a researcher or professional designs an architecture and wishes to understand the potential applications of this new design. Our pipeline can be utilized to retrieve similar models based on the architecture and obtain the performance scores of these retrieved similar neural networks. In the appendix, we provide a potential application. This demonstrates how users can retrieve similar architectures to determine the potential tasks for which their design could be applied or studied.
>
> - **Aid in Professional and Academic Search**
> Architecture search also benefits the retrieval of related classifications, aiding in finding detailed tags for professionals and researchers associated with these similar architectures. Examples of such tags include 'FLOPs,' 'model size,' 'model parameters,' and 'operator statistics distribution.' These benefits are crucial in modifying designs. It is helpful in many professional fields where a deep understanding of architecture is essential. (Consider how an image-based Google search returns a list of tags at the top as hits to narrow down the search purpose.)
>
>
> - **Architecture Data Modeling and Storage.**
> Pre-training architecture embeddings process serves as a new data storage format, allowing us to
> perform architecture manipulation similar to SQL schema. For instance, users can manage architectures via “group by”
> clause which clusters similar architectures together via measuring similarity among pretrained embeddings.
>
>
> - **Architecture Integration.** With both retrieval and storage in NAR, architecture integration can be achieved. A
> new architecture database can be built up to meet personal
> ized requirements through collaborating with other applications. For example, in recommendation systems, subscribed
> users will receive push notification for the latest architecture
> innovations, based on their specific interests.
>
>
>
> **Q2** "Some papers[1-3] ... should be discussed."
>
>
> **A2** We are truly thankful for the list of papers provided. These works are insightful and valuable. We will cite these papers in the revised version of our paper. Below is a comparison discussion:
>
> - [1] To obtain topological embeddings of neural architectures, it relies on auxiliary information such as the numbers of layers, channel expansion ratios, and kernel sizes. However, these details may not provide a comprehensive and precise description of the architecture. In contrast, our approach takes a more direct route by utilizing the graph structure itself to describe each motif in a neural network. This method avoids the limitations of incomplete auxiliary information, providing a more accurate and detailed representation of the architecture's topological characteristics.
>
> - [2] This paper proposes to tackle graph link prediction via a meta-learning framework to select model with better performances on unseen graph. one of the concerns in METAGL is the lack of modeling node features. Different from the graph learning in METAGL, our work takes architectures as graphs and operations as nodes, which enables us to perform feature extraction of architectures from a perspective of motifs. Thus, this method provides a more reliable representation learning of network architectures, involving operations, connections, and motifs.
>
> - [3] The  paper introduces RGCF, a method using meta-learning for efficient retrieval of GNN architectures in collaborative filtering, focusing on architectural search and rapid adaptation to new scenarios. In contrast, our paper proposes a unique approach for Neural Architecture Retrieval, employing contrastive learning to cluster neural architectures based on design similarities, and focusing on identifying connections between different designs. Our method utilizes the computational graph structure of neural architectures, dividing them into motifs for a more accurate and detailed representation, totally distinct from the RGCF's reliance on meta-features and task-level data augmentation.
>
> [1] Task-Adaptive Neural Network Search with Meta-Contrastive Learning. NeurIPS 2021.
>
> [2] MetaGL: Evaluation-Free Selection of Graph Learning Models via Meta-Learning. ICLR 2023
>
> [3] Retrieving GNN Architecture for Collaborative Filtering. CIKM 2023

---

> > ### Author Response · Authors · 2023-11-15
> > **Response to Reviewer NBZG**
> >
> > **Q3** "Another question is ... Chatgpt or other language-based retrieval model."
> >
> > **A3** We are grateful for your analysis. We add GPT-4 retrieval failed cases at the Appendix. Here we summarise as:
> >
> > (1). Most text descriptions of code do not include precise architectural information. For example, we copy the text descriptions of 'Falcom/animal-classifier' from the repository hub and feed them to GPT for retrieving similar architectures. However, GPT analysis fails and returns the steps of how to find similar models as follows: "Visit hub" -> "Search for similar models" -> "Filter results" -> "Review description" -> "Test Models" -> "Check for updates" -> "Read documentation and feedback", which are incapable of returning related architectures. The responses are demonstrated in Fig. 9(a) of [1].
> >
> > (2). In terms of feeding "code by language mode", GPT-4 is also incapable of handling the NAR task:
> >
> > - The code snippet can't represent the architecture because it may not reveal which architecture to load. For example, as shown in Fig. 9 (b) of [1], querying GPT-4 for the backbone code of Toolformer cannot retrieve similar architectures, but only return general concepts of natural language model.
> >
> > - The generative process of GPT is probabilistic in nature, leading to its inability to consistently retrieve accurate and stable similar architectures. For example, we query GPT4 three times for the same architecture "facebook/detr-resnet-50" by feeding with code snippet. As demonstrated in Fig. 10 (a)(b)(c) of [1], it give totally different generated answers.
> > Additionally, For the first two responses, GPT could only give a general concepts rather than specific models.
> >
> > - In many cases [2, 3], the model lacks an associated code repository, and the only way to get the architecture information is loading the checkpoint file and parsing the computational operators by tracing the forward process.
> >
> > [1] https://docs.google.com/document/d/1kKNq7K9asIlt89tE5Fl_oiq9xxilofuOU3wj83nEsaI/edit?usp=sharing
> >
> > [2] https://huggingface.co/jasmeen/dogs
> >
> > [3] https://huggingface.co/NimaBoscarino/dog_food
> >
> > **Q4** "One small question...describe it in detail?"
> >
> > **A4** We express our sincere gratitude for your detailed and insightful review. To clarify, the ground truth comprises the fine-grained classes demonstrated in Fig 4 (left and right bars). We describe the ground truth on page 6 in the 'Data Processing' section,  as follows: 'The key hints... are treated as the ground truth label of the neural network architecture.'

---

### Meta-Review · Area_Chair_mDXN · 2023-12-06

**Metareview:**

This paper introduce a new task named as Neural Architecture Retrieval (NAR) aiming to obtain a set of similar neural architectures given a query neural architecture. The authors leverages the motifs of neural architectures and uses two-level pretrain task to train the architecture representation for retrieval. A large new NAS dataset of 12k real-world neural architectures is also introduced. The reviewers acknowledge the technical soundness and its novelty and also point out some limitations, e.g., how to capture the relationship of architecture topology and architecture performance on multiple tasks. The rebuttal resolves most of the concerns and I opt for acceptance by considering the reviews and rebuttal.

**Justification For Why Not Higher Score:**

Please the meta-review.

**Justification For Why Not Lower Score:**

Please see the meta-review

---

### Decision · Program_Chairs · 2024-01-16

Accept (poster)